



# Characterizing the Spatial and Temporal Availability of Very High Resolution Satellite Imagery for Monitoring Applications

Myroslava Lesiv[1], Linda See[1], Juan Carlos Laso Bayas[1], Tobias Sturn[1], Dmitry Schepaschenko[1], Matthias Karner[1], Inian Moorthy[1], Ian McCallum[1] and Steffen Fritz[1]

[1]Ecosystems Services and Management Program, International Institute for Applied Systems Analysis (IIASA), Laxenburg A-2361, Austria

**Abstract.** Very high resolution (VHR) satellite imagery from Google Earth and Bing Maps is increasingly being used in a variety of applications from computer sciences to arts and humanities. In the field of remote sensing, this imagery is used to create detailed time-sensitive maps, e.g. for emergency response purposes, or to validate coarser resolution products such as global land cover maps. However, little is known about where VHR satellite imagery exists globally or the dates of the imagery. Here we present a global snapshot of the spatial and temporal distribution of VHR satellite imagery in Google Earth and Bing Maps. The results show an uneven availability globally, with biases in certain areas such as the USA, Europe and India. We also show that the availability of VHR imagery is currently not adequate for monitoring protected areas and deforestation, but is better suited for monitoring changes in cropland or urban areas. Supplementary data are available at https://doi.org/10.1594/PANGAEA.885767.

## 1 Introduction

Google Earth and Bing Maps provide visual access to VHR satellite imagery, defined here as imagery with a spatial resolution finer than 5m. We have gained new perspectives on our surroundings, both locally and internationally, as satellite imagery has become mainstream. We have also started to see this imagery being used across many different disciplines with increasing frequency. For example, using the search terms "Google Earth" or "Bing Imagery" in Scopus, which is a database of scientific abstracts and citations, reveals a steady increase from 2005 to 2016 in the number of papers that mention or use such imagery (Fig. S1), both across general domains (Fig. S2) and more specifically in remote sensing (Fig. S3). The imagery is used for different purposes but in remote sensing, mapping is the most frequent thematic area (Fig. S4) and map validation is the most commonly found application, i.e. producing an accuracy assessment of a map (Fig. S5). As many detailed features and objects can be seen from VHR imagery, e.g. buildings, roads and individual trees, reference data sets for map validation are increasingly being augmented with visual interpretation of Google Earth imagery, e.g.(Biradar et al., 2009; Bontemps et al., 2011). At the same time, applications such as Geo-Wiki (https://geo-wiki.org/) are using crowdsourcing to gather reference data sets for hybrid land cover map development and validation tasks based on visual interpretation of Google Earth and Bing Maps(Fritz et al., 2012, 2017, Schepaschenko et al., 2017, 2015, See et al., 2015a, 2016) while the Collect Earth tool (http://www.openforis.org/tools/collect-earth.html) uses Google Earth imagery to gather data for forest inventories (Bey et al., 2016).

VHR imagery is also extremely useful for a range of different environmental monitoring applications, from detecting deforestation to monitoring cropland expansion or abandonment. Unlike Bing Maps, Google Earth provides access to historical imagery, archiving the images as they are added to their system. This historical



imagery represents a valuable source of information for monitoring changes in the landscape over time. However, since Google Earth and Bing Maps present the satellite imagery in a seamless fashion, this may lead to the perception that the satellite data are continuous and homogeneous in nature, both in time and space. Yet in reality, the information is actually a mosaic of many images from different time periods, different spatial resolutions (15 m to 10 cm) and multiple image providers (from NASA's Landsat satellites to commercial providers such as DigitalGlobe); see e.g.(Microsoft, 2017). Hence the satellite image landscape is actually fractured, with much of the globe still covered by Landsat resolution imagery, i.e. 15 m when pan-sharpened. Although the European Space Agency's (ESA's) Sentinel-2 is now freely available and may slowly replace the base Landsat imagery in Google Earth, a 10 m spatial resolution is still not sufficient for visual interpretation of many landscape features. Moreover, for users of Google Earth and Bing Maps, little is known about the spatial availability of the VHR imagery or how much historical imagery exists in Google Earth and where it can be found, which can limit the use of this resource for environmental monitoring applications.

In this paper we provide a snapshot of the availability of VHR imagery globally by creating a systematic sample at each latitude/longitude intersection and extracting the type of imagery and the dates available for both Google Earth and Bing Maps. As mentioned above, we define VHR imagery as any imagery that has a spatial resolution finer than 5 m. Although the term 'VHR imagery' is often used to denote imagery at a resolution measured in centimeters, there are also other types of imagery available such as SPOT (1.5 to 5 m resolution), which can be useful in recognizing certain landscape features. This is the first time that metadata on the availability of VHR imagery in space and time has been made available for Google Earth and Bing Maps. The information can be used, for example, in the design of reference databases for remote sensing, particularly in applications that involve change detection. The snapshot provided here corresponds to the first week of January 2017, after which Google deprecated the Google Earth API/plugin and it was no longer possible to obtain the image dates from this source. With a focus on specific geographical areas, we then examine the availability of VHR imagery and its potential impact on monitoring world protected areas, deforestation, cropland and urban expansion.

## 2 Methods

### 2.1 Data extraction from Google Earth and Bing Maps

The dates of the images were extracted from Google Earth and Bing Maps using the API provided by each application on a systematic grid with a spacing of 1 degree or circa 100 km at the equator placed over land surface areas of the Earth. The Google Earth API was deprecated on 11 January 2017 so the Google Earth historical imagery dates were extracted just prior to this deprecation. The Bing Maps dates were extracted at the same time. For Bing Maps, only one satellite image is available at each location while Google Earth has historical imagery so the dates of all historical imagery were recorded at each grid point.

### 2.2 Spatial-temporal patterns of the image dates

The spatial distribution of the image dates from Bing Maps and Google Earth was plotted globally. For Bing Maps this corresponds to the imagery available as of 11 Jan 2017 while for Google Earth, this was the most recent date at each location. A comparison between Bing Maps and Google Earth was then made showing different attributes including: (i) the locations where there is no difference in the image dates between Bing Maps and Google Earth;



(ii) the locations where either Bing Maps or Google Earth has the most recent imagery; and (iii) the locations where only Bing Maps or Google Earth are available.

A number of additional maps were plotted for Google Earth imagery because of the availability of the historical imagery. The first is the number of historical images available in Google Earth, which shows those regions with

abundant time series and those with a lack of historical information. The vector of imagery dates at each location were then queried to extract the following indicators:

- Number of unique years: some of the historical imagery in Google Earth is concentrated in the same year so this indicator shows how many unique years are contained within the Google Earth archive, which is relevant information for monitoring change over time;

- An index of gaps in the time series: The number of unique images at a location was divided by the difference in years between the oldest and most recent image. Values of 1 indicate no time gaps in the time series; as the index decreases, the gap in the time series increases. This indicator was calculated for the USA and India as both countries have high numbers of images and high numbers of unique images.

- Number of seasons: the dates were grouped by the four seasons of winter (December, January, February),

spring (March, April, may), summer (June, July, August) and autumn (September, October, November); this indicator shows the number of historical images that fall in each of the four seasons, which is a relevant indicator for landscapes that change seasonally.

The image dates were then summarized by region, i.e. at the sub-continental level (Fig. S7), to calculate the percentage of grid points containing VHR imagery and the recent date occurring most frequently in these regions.

For Google Earth, additional indicators were calculated including the average number of images per grid point, the average number of unique years per grid point and the average difference between the oldest and the most recent date.

Finally, the Pearson correlation coefficients between the number of images available in Google Earth and population density (as a proxy for urban areas) was calculated (Table S1) where the number of images in each

250 $m^2$ grid cell was extracted. Population density was obtained from the Global Human Settlement Population Grid for the year 2015 and has been produced by the Joint Research Centre of the European Commission(JRC, 2015). The idea was to determine if there is a bias in the amount of VHR imagery in urban areas.

### 2.3 Availability of VHR images in protected areas

The World Protected Areas data set from UNEP-WCMC(Juffe-Bignoli et al., 2014) contains the boundaries of

protected areas globally. This layer was used to extract those sample points that fell within protected areas of all categories (from most to least protected), which were then disaggregated by major world region. The percentage of sample points with VHR imagery in Google Earth and Bing Maps was then calculated along with the median of the date of the imagery in Bing Maps and the most recent and oldest dates in Google Earth.  The average number of images in Google Earth was then calculated by region along with the number of unique years and the

average number of seasons per location.



### 2.4 Availability of VHR images in areas with high rates of deforestation

To examine the availability of VHR images in areas with high deforestation, we selected regions that have the highest forest cover change according to FAO's Global Forest Resources Assessment in 2015(FAO, 2016). In particular, we chose:

1. Regions where crop expansion is the main driver of forest loss, i.e. the Amazon, the Congo basin, Indonesia and Malaysia, and Laos and Cambodia; and

2. Developed countries with intensive forest management: i.e. Sweden and Finland.

The sample points falling in the regions listed above were then extracted from the full data set. A forest mask(Schepaschenko et al., 2013) was used to determine the number of sample points that fall within forest areas.

We then calculated the percentage of VHR images in Google Earth and Bing Maps by region in forest areas along with the most frequent year of the imagery for Bing Maps, the most frequent oldest and most recent imagery in Google Earth as well as the average and unique number of images in Google Earth.

### 2.5 Availability of VHR images in areas with cropland

Visual interpretation of VHR imagery in the context of cropland can differ based on whether the image falls inside

or outside of a growing season. We used the MEaSUREs Vegetation Index and Phenology (VIP) Global Data Set, produced by NASA(Didan and Barreto, 2016), which contains information on a range of different phenological metrics at a 0.05 degree resolution. The relevant measures extracted from this product at the sample locations included the number of growing seasons and their start and end dates. We then compared the dates of the imagery with the growing season dates to determine if the imagery at a sample location falls in or outside of a growing

season or whether imagery is available for both cases. This information is relevant for applications related to cropland monitoring, where image interpretation would benefit from having scenes both inside and outside of a growing season.

We then selected a list of countries to examine the availability of VHR images in cropland areas in more detail. The criteria used for selection were the following:

1. Countries with poor cropland monitoring systems, identified as countries with the highest food security risks(See et al., 2015b), i.e. Angola, Chad, Ethiopia, Mongolia, Mozambique and Namibia;

2. Countries with a large cropland expansion or cropland loss since 2000, i.e. Nigeria, Indonesia, Brazil, Argentina, Tanzania, Australia, India and Sudan based on FAO statistics and a recent study on risks to biodiversity due to cropland expansion and intensification(Kehoe et al., 2017);

3. The USA, which was chosen because it has the best coverage of VHR imagery.

The sample points falling in the countries listed above were then extracted from the full data set and divided into two subsets based on whether the points fall inside or outside of a cropland area. The Unified Cropland Layer produced for global agricultural monitoring at a resolution of 250 m(Waldner et al., 2016) was used as a cropland mask to differentiate between areas of cropland presence or absence. The number of locations with VHR images

was then calculated along with the most frequent year in both Google Earth and Bing Maps, as well as the total and unique number of images in Google Earth.



### 2.6 Availability of VHR images in urban areas

Using the layer of urban and rural areas developed by the Joint Research Center (JRC) at a 1 km² resolution(JRC, 2016), the number of locations with VHR images falling in these two classes was calculated, along with the average number of unique years in Google Earth, the most frequent oldest and most recent images in Google Earth

and the most frequent year in Bing Maps.

### 2.7 Software used

All of the analysis in the paper has been done in the R statistical environment. All the figures were produced using ESRI's ArcMap v.10.1 GIS software.

### 3 Results

### 3.1 Spatial-temporal distribution of VHR satellite images

Fig. 1 and 2 show the spatial distribution of the most recent dates of VHR satellite imagery available in Bing Maps and Google Earth, respectively, while Fig. 3 provides a comparison of the two together.

The most recent imagery in Bing Maps (Fig. 1) from 2016 can be found in eastern Canada, some cities in Australia, India, Nepal and Bangladesh. Imagery from 2014 and 2015 is predominantly found in Australia and India while

there is a reasonable coverage of the rest of the world with images from 2009 to 2013. Bing Maps are not available in much of Indonesia and Papua New Guinea, eastern China, some of the Congo Basin and at northern latitudes. Some older imagery (2002 to 2008) can be found in parts of Brazil and neighboring countries.

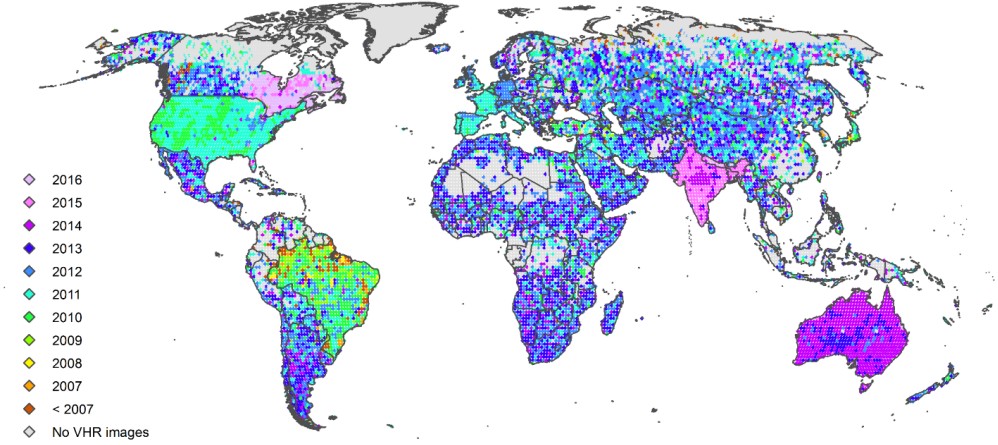

**Figure 1: The dates of the most recent VHR satellite imagery (< 5 m resolution) available in Bing Maps as of January**
**2017 (Software: Esri®ArcMap™ 10.1)**

In contrast, imagery from Google Earth is generally more recent than Bing imagery. In Google Earth (Fig. 2), continuous areas of very recent imagery (2016) can be found in India, parts of South America and some African countries. There is a noticeable lack of VHR imagery in the northern latitudes, parts of the Amazon and desert areas. This is particularly evident for Australia, where Bing Maps is either the only source of VHR imagery or




contains the most recent imagery (Fig. 3). The coverage of the Amazon is also much better in Bing Maps than it is in Google Earth.

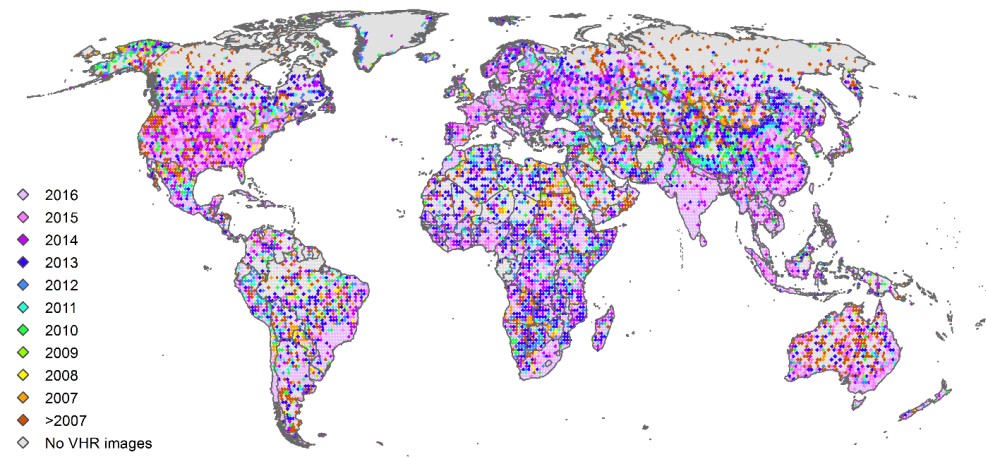

**Figure 2: The dates of the most recent VHR satellite imagery (< 5 m resolution) available in Google Earth as of January 2017 (Software: Esri®ArcMap™ 10.1)**

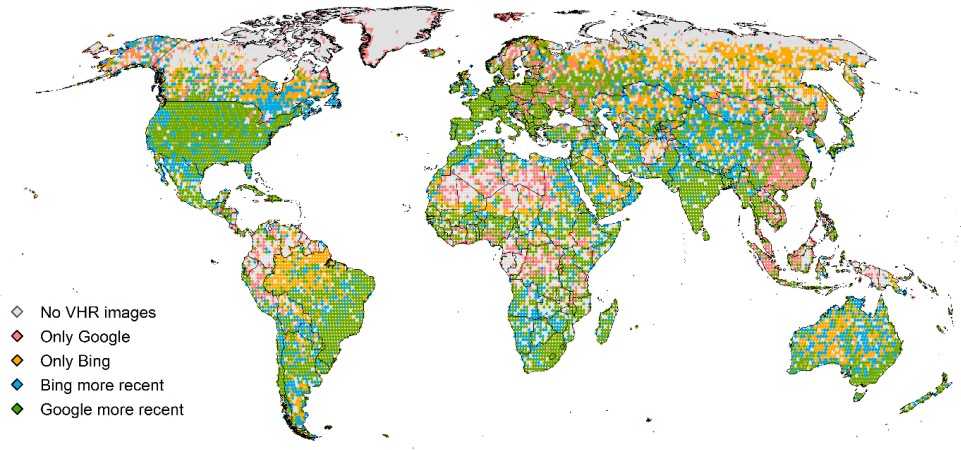

**Figure 3: Comparison of the most recent VHR satellite imagery (< 5 m resolution) available in Bing Maps and Google Earth as of January 2017 (Software: Esri®ArcMap™ 10.1)**

A regional analysis of the availability of VHR imagery in both Google Earth and Bing Maps is provided in Table 1. The coverage of VHR imagery is very high for both Google Earth and Bing Maps for Central America, Southern and Western Europe, and Eastern and Southern Africa but as observed previously from Fig. 1 and 2, Google Earth tends to be more recent than Bing Maps, i.e. 2016 for Google Earth versus 2011/2013 for Bing Maps. Similar high coverage can be found in Eastern and Southern Asia for both, but the dates of Google Earth and Bing Maps are similar.





The results also confirm the previous findings that VHR imagery is available for almost all of Australia and New Zealand when considering Bing Maps while the coverage is lower (70%) for Google Earth; the dates are also similar although slightly more recent for Google Earth. Bing Maps coverage is also better than Google Earth for South America, and Western and Central Asia.

5   The worst coverage can be found in North America, where only half the area is covered by VHR imagery in both Google Earth and Bing Maps, and Eastern Europe, where Google Earth has only 39% VHR imagery and Bing Maps has 58%. This is probably due to the fact that these regions cover high northern latitudes, where there is no availability of VHR imagery (Fig. 3). In contrast, Google Earth has better coverage in Northern European, Southeastern Asia and Middle Africa compared to Bing Maps.

10   Overall, Google imagery is more recent than Bing Maps but Bing Maps provide the spatial complementarity to Google in South America, Australia, New Zealand and the northern part of Eastern Europe. North America, Southern Europe, Southern Africa, and Southern and Southeastern Asia have the richest archive of images, while Eastern and Northern Europe, Central Asia, Northern and Central Africa have mostly only one or two images per location.



**Table 1: The availability of VHR satellite imagery (< 5 m resolution) from Google Earth and Bing Maps by region**

| Region | Total number of grid points | Google Earth | | | | | Bing Maps | |
|---|---|---|---|---|---|---|---|---|
| | | Coverage with VHR imagery (%) | Average number of images per grid point | Average number of unique years per grid point | Average difference between the oldest and the most recent date (years) | Most recent year, calculated as the median | Coverage with VHR imagery (%) | Most recent year, calculated as median |
| North America | 3421 | 47% | 5 | 5 | 10 | 2016 | 51% | 2011 |
| Central America | 219 | 95% | 4 | 3 | 6 | 2016 | 94% | 2013 |
| Most South America | 1534 | 72% | 4 | 3 | 5 | 2016 | 88% | 2010 |
| Northern Europe | 318 | 90% | 3 | 2 | 3 | 2015 | 64% | 2012 |
| Southern Europe | 142 | 99% | 5 | 4 | 8 | 2016 | 96% | 2011 |
| Western Europe | 136 | 99% | 4 | 3 | 7 | 2016 | 100% | 2012 |
| Eastern Europe | 3129 | 39% | 3 | 2 | 4 | 2016 | 58% | 2012 |
| Northern Africa | 740 | 73% | 3 | 2 | 4 | 2013 | 67% | 2013 |
| Western Africa | 509 | 70% | 3 | 3 | 4 | 2016 | 67% | 2013 |
| Middle Africa | 536 | 75% | 2 | 2 | 3 | 2013 | 65% | 2013 |
| Eastern Africa | 530 | 96% | 4 | 3 | 5 | 2016 | 90% | 2013 |
| Southern Africa | 243 | 100% | 5 | 3 | 7 | 2016 | 99% | 2013 |
| Eastern Asia | 1212 | 89% | 3 | 3 | 4 | 2013 | 82% | 2012 |
| Western Asia | 415 | 79% | 4 | 3 | 5 | 2016 | 93% | 2013 |
| Southern Asia | 583 | 91% | 7 | 4 | 8 | 2016 | 92% | 2015 |
| Southeastern Asia | 372 | 89% | 5 | 3 | 6 | 2016 | 58% | 2013 |



| | | | | | | | | |
|---|---|---|---|---|---|---|---|---|
| Central Asia | 468 | 65% | 2 | 2 | 3 | 2016 | 90% | 2012 |
| Australia and New Zealand | 727 | 70% | 4 | 3 | 5 | 2015 | 100% | 2014 |



As some of the historical images are from the same year, Fig. 4 shows the number of unique years for which VHR imagery is available in Google Earth. The areas with the most imagery available are the USA, India, parts of Eastern Europe and Indonesia, and some of the more populated regions across all the continents, e.g. the southern part of Brazil, the eastern coast of Australia and the southeastern part of South Africa. Overall, the majority of the

5 world is covered by only 1 to 3 images per location, which may explain why there is only a medium correlation between population density and the number of images in the regions of Northern Africa (r=0.46, p-values<0.001), South America (r=0.40, p-values<0.001), Western Asia (r=0.38, p-values<0.001), and Eastern Europe (r=0.29, p-values<0.001) (Table S1) and low or no correlations in the rest of the world. Similar spatial patterns were found when plotting the total number of VHR images available in Google Earth (Fig. S8).

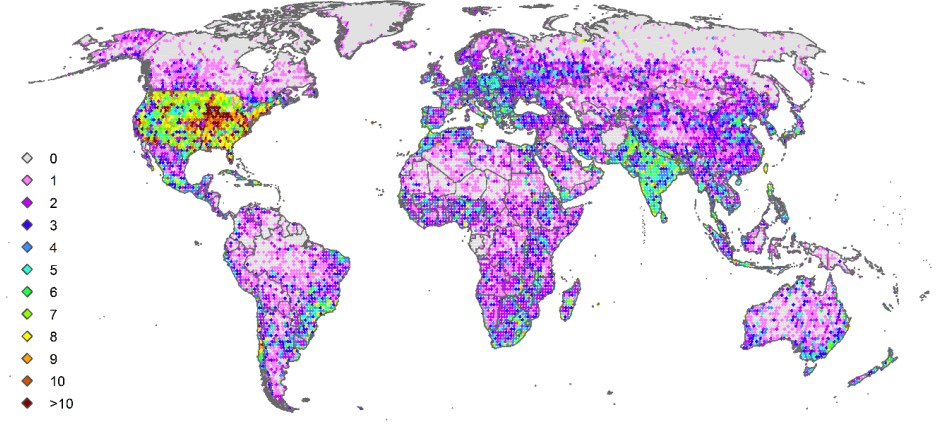

**Figure 4: The number of unique years of VHR satellite images (< 5 m resolution) available in Google Earth (Software: Esri®ArcMap™ 10.1)**

Seasonal patterns are also evident in the historical archive of Google Earth. Fig. 5 shows the availability of VHR imagery according to the number of seasons represented. Very few places have imagery from all 4 seasons (winter,

15 spring, summer and autumn) while 3 seasons are available in majority of the USA, India and Eastern Europe, mirroring the pattern found for the number of images.



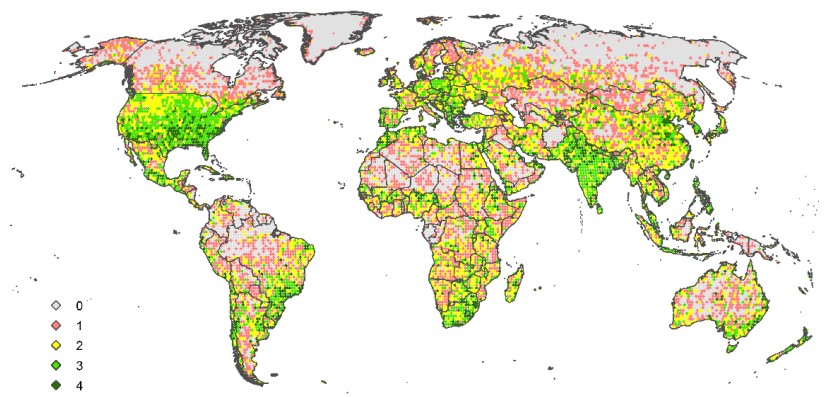

**Figure 5: The availability of VHR imagery (< 5 m) in Google Earth based on the number of seasons represented by the dates (Software: Esri®ArcMap™ 10.1)**

We now examine the availability of VHR imagery in relation to four domains where such imagery has value for environmental monitoring, i.e. monitoring of protected areas; monitoring of areas that have high rates of deforestation; monitoring areas with cropland, where the latter application has relevance for food security; and monitoring urban areas.

### 3.2 Availability of very VHR images for monitoring protected areas

Table 2 summarizes the availability of VHR images inside protected areas(Juffe-Bignoli et al., 2014) by major world region. Greenland has been excluded from this analysis due to the absence of VHR images in this area. The coverage of protected areas in Bing Maps is better than Google Earth for most regions except for Africa and Western, Southern and Northern Europe, where it is only slightly lower. The comprehensive coverage by Bing Maps in Australia and New Zealand is again evident when compared to Google Earth while coverage in South America and Eastern Europe are considerably lower in Google Earth compared to Bing Maps. Google Earth images are generally more up-to-date than Bing Maps, have an average of at least 3 images per location, and cover at least 2 different seasons.



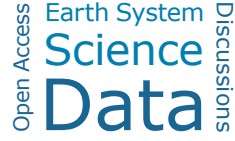

**Table 2: Availability of VHR imagery ($<$ 5 m) inside protected areas grouped by regions**

| Regions | Total number of points inside protected areas | Google Earth | | | | | | Bing Maps | |
| --- | --- | --- | --- | --- | --- | --- | --- | --- | --- |
| | | Coverage with VHR imagery (%) | Most recent year, calculated as the median | Oldest year, calculated as the median | Average number of images per location | Average number of unique years per location | Average number of different seasons per location | Coverage with VHR imagery (%) | Most recent year, calculated as the median |
| Northern America and Central America (excl. Greenland) | 361 | 54% | 2013 | 2006 | 4 | 3 | 2 | 59% | 2011 |
| South America | 322 | 48% | 2013 | 2009 | 2 | 2 | 2 | 84% | 2010 |
| Australia and New Zealand | 117 | 63% | 2014 | 2006 | 3 | 3 | 2 | 100% | 2014 |
| Asia (Southern, Eastern, Southern-Eastern, Central and Western) | 337 | 72% | 2013 | 2009 | 3 | 2 | 2 | 87% | 2012 |
| Africa | 324 | 84% | 2013 | 2009 | 3 | 3 | 2 | 79% | 2013 |
| Western, Southern and Northern Europe | 116 | 90% | 2014 | 2010 | 3 | 3 | 2 | 83% | 2012 |
| Eastern Europe | 349 | 35% | 2014 | 2009 | 3 | 3 | 2 | 58% | 2012 |



### 3.3 Availability of VHR images for monitoring deforestation

Table 3 illustrates the availability of VHR imagery within selected regions that have the highest forest cover change based on FAO's Global Forest Resources Assessment 2015(FAO, 2016). There is good coverage by Bing Maps in the Amazon and the Congo basin but there is only one image available and the most recent, frequent year found is 4 to 6 years old. In contrast, Google Earth has relatively poor coverage in the Amazon and only 1 image available on average with similar years. For the Congo basin, the coverage is better but still poorer than Bing Maps and only 1 unique image is available on average. For the other regions, the availability of Google Earth VHR imagery is quite good although only 2 unique years are available on average and the images are more recent than Bing Maps.

**Table 3. Availability of VHR imagery (< 5 m) in areas with the highest forest cover change**

| Region | Number of points | Number of points classed as 'forest' | Google Earth | | | | | Bing Maps | |
|---|---|---|---|---|---|---|---|---|---|
| | | | Coverage by VHR imagery (%) | Oldest year, calculated as the median | Most recent year, calculated as the median | Average number of images | Average number of unique years | Coverage by VHR imagery (%) | Most frequent year, calculated as the median |
| Amazon biome | 694 | 601 | 48% | 2010 | 2012 | 1 | 1 | 93% | 2010 |
| Congo basin | 439 | 277 | 74% | 2012 | 2013 | 2 | 1 | 92% | 2012 |
| Indonesia and Malaysia | 182 | 171 | 82% | 2011 | 2015 | 3 | 2 | 47% | 2012 |
| Laos and Cambodia | 34 | 28 | 85% | 2011 | 2014 | 2 | 2 | 78% | 2012 |
| Sweden and Finland | 142 | 123 | 85% | 2012 | 2014 | 2 | 2 | 67% | 2012 |

### 3.4 Availability of VHR imagery in cropland areas

To monitor cropland, particularly the presence of annual crops that can appear quite differently on satellite imagery depending on the growing season, it is useful to know the availability of VHR imagery both inside and outside of a growing season, which is shown in Fig. 6. The distribution shows that most of the images are either taken during the growing season or there is imagery available both inside and outside of this period. Areas with imagery available only outside of the growing season can be found in the transition zones between desert and agricultural areas in the Sahel, and in the desert areas of Australia and western China, where there is less agriculture.





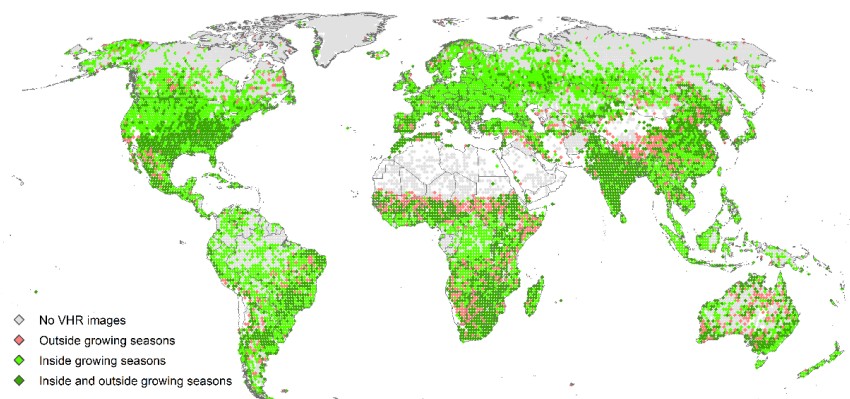

**Figure 6: The distribution of VHR imagery (< 5 m) in Google Earth based on the availability in and outside of a growing season. Areas in white, e.g. in the Sahara desert, were excluded since there is no data available due to absence of cropland (Software: Esri®ArcMap™ 10.1)**

5   For the countries selected as having either poor cropland monitoring or large expansion or loss of cropland since 2000 (see Methods), the availability of VHR imagery is shown in Table 4. The USA is also added as a contrast since it has large areas of cropland and good availability of VHR imagery (Fig. 6). The results show that the cropland areas in these countries are covered by more than 90% VHR imagery in Google Earth; there are similar findings in Bing Maps except for Nigeria and Indonesia, which still have high coverage. The only country for

10   which no VHR imagery is available is Mongolia, which is unsurprising given its location in the high northern latitudes where minimal VHR imagery tends to be available. Table S2 also shows that in some countries such as Ethiopia, Namibia, Nigeria, Indonesia, Tanzania and Australia, there are more images available in cropland versus non-cropland areas.

Bing Maps are generally older than Google Earth's most recent imagery but all countries have 2 or more historical

15   images available in Google Earth; some countries even have 5 or more images available, which span more than one season in a given year. Both the USA and India have the most images available in cropland areas although images from more unique years are available for the USA.



**Table 4: Availability of VHR imagery (< 5 m) inside cropland areas for selected countries**

| Selected countries | Google Earth | | | | | | | | Bing Maps | |
|---|---|---|---|---|---|---|---|---|---|---|
| | Total number of points | Number of points classed as 'cropland' | Coverage with VHR imagery (%) | Oldest year, calculated as the median | Most recent year, calculated as the median | Number of images | Average number of unique years | Average number of different seasons per location | Coverage with VHR imagery (%) | Most frequent year, calculated as the median |
| Angola | 102 | 13 | 100% | 2007 | 2013 | 2 | 2 | 92% | 2012 | 2 |
| Chad | 105 | 12 | 92% | 2010 | 2013 | 2 | 2 | 100% | 2013 | 1 (non-growing season) |
| Ethiopia | 91 | 33 | 94% | 2012 | 2016 | 6 | 3 | 94% | 2012 | 2 |
| Mongolia | 183 | 0 | 0 | - | - | 0 | 0 | 0 | - | - |
| Mozambique | 68 | 4 | 100% | 2010 | 2013 | 2 | 2 | 100% | 2012 | 2 |
| Namibia | 75 | 3 | 100% | 2005 | 2016 | 5 | 3 | 100% | 2012 | 2 |
| Nigeria | 74 | 53 | 100% | 2007 | 2016 | 6 | 4 | 87% | 2012 | 2 |
| Indonesia | 153 | 23 | 96% | 2007 | 2016 | 7 | 4 | 74% | 2012 | 1 (growing season) |
| Brazil | 705 | 52 | 92% | 2005 | 2014 | 4 | 3 | 100% | 2011 | 2 |
| Argentina | 284 | 60 | 93% | 2004 | 2016 | 4 | 3 | 100% | 2013 | 2 |
| Tanzania | 77 | 26 | 100% | 2008 | 2015 | 4 | 3 | 96% | 2012 | 2 |
| Australia | 699 | 45 | 98% | 2010 | 2015 | 4 | 3 | 100% | 2014 | 2 |

Top countries with highest priority to map croplands — Top countries with cropland expansion, 2000-2014 — Top countr...

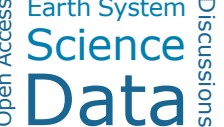

|  |  |  |  |  |  |  |  |  |  |  |
|---|---|---|---|---|---|---|---|---|---|---|
| India | 274 | 170 | 100% | 2004 | 2016 | 9 | 5 | 100% | 2015 | 2 |
| Sudan | 209 | 29 | 98% | 2009 | 2015 | 3 | 2 | 100% | 2012 | 2 |
| USA | 1108 | 172 | 99% | 1994 | 2015 | 10 | 9 | 2 | 100% | 2011 |



### 3.6 Availability of VHR imagery for monitoring urban areas

Table 5 presents the distribution of sample locations that fall within urban and rural areas(JRC, 2016); the majority of sample points fall outside of these two classes in unpopulated areas and are not shown here. Of those falling in urban areas, coverage is 100% in Google Earth and still high in Bing Maps (87%). In rural areas the coverage is

5 lower but nevertheless good at around 80% for both Google Earth and Bing Maps. For urban areas, the number of unique years is 6, with a broad range of older and more recent imagery in Google Earth. Hence it is possible to use the imagery for some change detection in urban areas or validation of remotely-sensed urban products. Bing Maps tend to be older than the most recent Google Earth imagery but may add additional information for change detection or validation purposes.

**Table 5: Availability of VHR imagery (< 5 m) in urban and rural areas**

| | Number of points | Google Earth | | | | Bing Maps | |
| | | Coverage by VHR imagery (%) | Average number of unique years | Oldest year, calculated as the median | Most recent year, calculated as the median image | Coverage by VHR imagery (%) | Most frequent year, calculated as the median |
|---|---|---|---|---|---|---|---|
| Urban areas | 219 | 100% | 6 | 2004 | 2016 | 87% | 2012 |
| Rural areas | 2790 | 79% | 4 | 2005 | 2014 | 82% | 2011 |

### 4 Discussion

The results have shown that there is clearly unequal spatial and temporal coverage by VHR imagery across the globe. There are parts of the world that have no VHR imagery, i.e. high northern latitudes, countries in the

15 northwestern part of South America, e.g. Ecuador and Colombia, parts of the Saharan Desert, parts of the Congo Basin and Indonesia/Papua New Guinea. Hence it is difficult to do any monitoring in these areas since there is only Landsat (pan-sharpened 15 m resolution) base imagery available. In the rest of the world there is some complementarity between Google Earth and Bing Maps, e.g. there are only Bing Maps present in parts of Canada, the Amazon, former Soviet Union countries and parts of Australia where Google Earth has no coverage. In



contrast, Google Earth imagery adds very little additional spatial coverage but tends to be more recent than Bing Maps and has the benefit of a historical archive, which adds potential value for change detection and monitoring purposes. However, the reality is that for applications where a time series of images would greatly benefit monitoring, the amount of historical imagery is actually quite small.

5       We then focused on four applications where the use of VHR satellite imagery would greatly benefit monitoring and change detection, i.e. protected, forested, cropland and urban areas. Due to increased competition for land(UNCCD, 2017), protected land areas are threatened, impacting biodiversity and natural resources(Costanza et al., 2014; Juffe-Bignoli et al., 2014); hence monitoring is vital. The availability of VHR imagery in protected areas was surprisingly poor in North America, Eastern Europe and South America, particularly in Google Earth

within the latter two regions. On average there are only 2 to 3 historical images in different years; hence monitoring is possible in some parts of the world but it is limited.

For deforestation, the picture is worse, particularly in a region such as the Amazon. Although coverage by Bing Maps is relatively good, less than 50% of the points falling in the Amazon biome were covered by VHR imagery in Google Earth, with on average only 1 year of imagery. Thus, there is a clear lack of information in the historical

archive for monitoring change. The spatial-temporal coverage is better for Indonesia and Malaysia where there are three images on average in different years in Google Earth while most of the other regions have 2 years on average. Although new tools and products for monitoring deforestation have appeared recently, e.g. through Global Forest Watch, the basis of change detection is Landsat imagery, which still requires validation with VHR imagery.

For studies in crop expansion or abandonment and urbanization, the availability of suitable VHR imagery is much better. The coverage by VHR imagery in countries with poor crop monitoring systems, those currently subject to cropland expansion and losses, and those areas classified as urban is extremely high. There are time series of images available, and for cropland, images from more than one season. Hence there is quite some potential for using this resource for change detection in cropland and urban areas and the validation of remotely-sensed

products.

From the Scopus search and the breakdown by discipline (Fig. S1 and S2), the increasing value of Google Earth and Bing Maps are evident. Fig. S1 and S5 confirm the increasing use of imagery from Google Earth and for validation tasks in remote sensing, respectively, while new crowdsourced reference data sets based on Google Earth and Bing Maps are appearing(Fritz et al., 2017; Laso Bayas et al., 2017). The collection of in-situ data is

resource intensive, both in terms of time and money, e.g. the LUCAS (Land Use Cover Area frame Survey) data set represents the only source of in-situ data for EU member countries where ca 300K points are surveyed on the



ground every 3 years(Gallego, 2011). The implementation in 2018 alone will cost more than 12 million Euros(Eiselt, n.d.). Hence the visual interpretation of VHR imagery (via Google Earth and Bing Maps) has become a more cost-effective approach for building reference data sets for the validation of land cover and land use maps, as well as inputs to the training algorithms that create these products. Hence from an environmental and research

perspective, it is important that access to these data sources continues and that gaps in VHR imagery are filled where possible. The costs of purchasing data from providers such as DigitalGlobe are prohibitive, particularly when applications are global or large areas need to be monitored. Moreover, we are increasingly moving away from the development of static products of land cover and land use and are interested in detecting change over time, e.g. forest loss and gain over time(Hansen et al., 2013) or monitoring the change in water bodies over a 32

year period(Pekel et al., 2016). Fig. S6 shows that the majority of papers are using imagery from different time periods, which reflects this trend. As new land cover products appear, e.g. the recent ESA CCI land cover time series from 1992-2012, access to VHR imagery for validation of land cover change is vital, particularly if users want to independently validate the product for their own user needs. The spatial-temporal metadata on the image dates and the availability of VHR imagery presented here can be used to guide sample design for validation of

land cover time series. However, this is only a snapshot in time so having a new API for accessing the dates of imagery in Google Earth as well as other meta-information about the satellite imagery would be extremely useful for a range of applications.

At the same time, there are encouraging initiatives to improve the availability and accessibility of VHR imagery, e.g. the satellite company Planet has 149 of their small dove satellites orbiting the Earth, which together provide

daily coverage of the Earth's land surface at a 3 to 3.5m resolution. Free access to 10,000 km2 of VHR imagery per month is available for non-commercial purposes (https://www.planet.com). The Radiant Earth initiative from the Bill and Melinda Gates Foundation and the Omidyar Network is making a considerable amount of satellite imagery free for humanitarian and environmental causes (https://www.radiant.earth). Most of the value in VHR satellite imagery is in the up-to-date nature of the information. Commercial image providers should be encouraged

to unlock their historical archives, where the information has much less commercial value, and share the imagery via applications such as Google Earth. Not only does this benefit research, it can aid environmental monitoring by many different stakeholders in the public sector as well as non-governmental organizations and charities. New applications can be built to mobilize citizens to aid in change detection, which can help tackle many pressing environmental problems. The value of VHR satellite imagery available through Google Earth and Bing Maps

should not be underestimated; it has the potential to be so much more.





## 5 Supplementary link

It will be included by Copernicus.

## 6 Data Availability

The data is available as a flat file (.tab) containing the latitude and longitude of each sample point, the year of the

Bing Maps image and the years available in the Google Earth imagery as of 10 January 2017. Here is the link to

the data: https://doi.org/10.1594/PANGAEA.885767.

## 7 Author Contributions

Myroslava Lesiv conceived the original idea and did the analysis in the paper. Linda See, Juan Carlos Laso Bayas, Dmitry Schepaschenko and Steffen Fritz provided suggestions for different analyses and wrote the paper with

Myroslava Lesiv. Juan Carlos Laso Bayas undertook the Scopus search and subsequent analysis. Tobias Sturn and Matthias Karner programmed the data collection from Google Earth and Bing Maps. Inian Moorthy and Ian McCallum provided recommendations for improvements to the paper and additional editing.

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
