# Peer review of "Characterizing the Spatial and Temporal Availability of Very High Resolution Satellite Imagery for Monitoring Applications"

_Earth System Science Data, 2018_

## Referee Comment (RC1) · Anonymous Referee #1 · 21 Apr 2018

Page 3, Line 15: Capitalize May Minor Detail, on Page 18, Line 18, Global Forest Watch is powered by Google Earth Engine, so maybe worth mentioning that tool, it also needs validation with VHR imagery when doing classifications.

Other than that, no big comments, this is very good article and very useful...I didn't realize the lack of literature in this area so thank you for doing this!

---

## Referee Comment (RC2) · Anonymous Referee #2 · 19 Jun 2018

Earth sciences researchers, at least those that publish wide ranges of data in ESSD, use imagery from Google Earth rarely (see below) and from Microsoft Bing almost never. This manuscript, which counts availability of Google Earth and Microsoft Bing scenes at various terrestrial locations of our planet, has minimal utility and relevance for ESSD readers. Separate from irrelevance, it fails substantially in the quality of its presentation.

If VHR scenes become more available and more useful, readers will need a much more organised, systematic and compelling guide than the one provided here. At best, it seems pre-mature and not up to the quality expected for ESSD.

Please also note the supplement to this comment:
https://www.earth-syst-sci-data-discuss.net/essd-2018-13/essd-2018-13-RC2-
supplement.pdf

**Supplement:**

Review ESSD-2018-13, VHR satellite imagery

Earth sciences researchers, at least those that publish wide ranges of data in ESSD, use imagery from Google Earth rarely (see below) and from Microsoft Bing almost never. This manuscript, which counts availability of Google Earth and Microsoft Bing scenes at various terrestrial locations of our planet, has minimal utility and relevance for ESSD readers. Separate from irrelevance, it fails substantially in the quality of its presentation.

Of ESSD data sets published to date, 12 used and cited Landsat imagery. One has already published a data set using ESA Sentinel-2 images. Six others presented .kml files to allow readers to browse data locations using Google Earth - a nice convenience used by many members and institutions among the data sharing community but a false positive when one does a word search for 'Google Earth'. Only one ESSD data set over 10 years actually used scenes from Google Earth, in their case to estimate water colour in un-sampled remote branches of Amazonian rivers (naming, by the way, Landsat as the original source). Those authors issued this clear explicit qualification: "It should be noted that both methods are qualitative and subject to error" (https://doi.org/10.5194/essd-8-651-2016). Readers can easily confirm far less than 1% usage of Google Earth scenes in ESSD data products (roughly 300 published in ten years) and a ratio of at least 10 to 1 of Landsat vs Google Earth. No ESSD papers mention Microsoft Bing. Note that these data providers and data users prefer the access and reliability of publicly-available data from USGS or ESA/Copernicus over the higher spatial resolutions (by definitions adopted by these authors, 5 metre vs 15 metre) offered by SPOT or commercial sources such as DigitalGlobe.

In Supplement Figure 2, showing data extracted from SCOPUS (why did these authors not use Web of Science / CLARIVATE instead, for much broader topic coverage?), the authors expect readers to accept that over more than 10 years (2005 to 2016) 1500 (26% of 5756 total publications) papers in the general field of Earth Sciences (probably we can ignore planetary sciences for purposes of assessing utility of earth images) and perhaps an additional 900 papers (16%) in a field called Environmental Sciences mentioned 'Google Earth' or 'Bing Imagery'. Out of 100s of thousands of papers published during those years, roughly 1000 (because the authors give us no breakdown, we assume half for Google Earth and half for Microsoft Bing) mentioned Google Earth? Far fewer - 96 from Figure S5 - mentioned Google Earth in conjunction with remote sensing. Subtracting those papers that referenced Google Earth in terms of a .kml file (as the ESSD experience suggests happens frequently), and a few others that used Google Earth scenes with explicit qualifications (again referencing the ESSD experience), something like 50 papers - 5 per year over 10 years - mentioned Google Earth in conjunction with remote sensing? From their own data, these authors demonstrate the minuscule impact of Google Earth scenes on earth system research (and their search does not distinguish lower resolution from higher resolution images)!

Google Earth and Microsoft Bing use multiple sources and elaborate processes to acquire, assemble and display their maps. Any individual scene thus carries a complicated, evolving and largely proprietary provenance, not openly documented and generally not available to researchers. Any Google Earth or Microsoft Bing scenes that display features of 1 to 5 metres in extent must have supplemented Landsat imagery with higher-resolution sources (listed above), but in a manner hidden to users. Although convenient, useful and fun (!) for a wide variety of personal navigation and information purposes, earth and environmental science researchers strongly prefer original aerial or satellite images because of their far-superior metadata, documentation, veracity, etc. If these authors had done a similar search on 'Landsat' or 'SPOT' or even 'Sentinel-2' (not launched until 2015) using SCOPUS, this reviewer predicts they would have generated tens of thousands of hits in a general search and hundreds to thousands if they looked again for remote sensing. Landsat and even Sentinel-2 dominate the image 'market' for research.

This reviewer finds a most-recent Google Earth image of my present location dated July 2014, with both Landsat and Copernicus (e.g. ESA Sentinel) attributed as sources (Sentinel for a 2014 image???), carrying a Google 2018 copyright notice. I estimate - because I can not determine from reliable metadata - a spatial resolution for urban features of 0.5 to 1 metre in 2014 which degrades rapidly and substantially (and occasionally loses registration) in prior images from 2011, 2009, 2005, etc. From USGS EarthExplorer I can relatively easily find mid-day Landsat 8 images

of my location, at resolutions of 30 (vis) or 15 (panchromatic) metres.  Applying the most stringent cloud removal filters (<10% for both scene and image) I can download 50 strikingly clear images covering all seasons from late 2013 up to last week, sizes 10MB to 1GB depending on my choices of format and on my network speeds.  I could access another 50 images with a slightly different but overlapping view angle.  For every image I could document and share: time, orbit, acquisition parameters, exact file name and format, etc.  If I want to monitor and describe annual or seasonal patterns of snow cover, surface hydrology (e.g. reservoir levels), vegetation changes, agricultural choices and practices, wildfire burn areas, urban development, etc., I will rely on these Landsat images in preference to Google Earth images of variable temporal availability and uncertain provenance.  If I considered cloud distribution and coverage as valid features of my research (to validate ground-based radiation measurements, for example), I could choose among hundreds of additional images over the same time period.  For most monitoring purposes, researchers need reliable time coverage while tending to avoid the huge file sizes associated with 0.5 metre spatial images if available.  The authors make much of their VHR definition, but for most sites and most usages that highest resolution applies only to most-recent scenes and - due to large files sizes already mentioned - proves impractical for compilation and use on personal computers?  Even the World Urban Data Access Portal Tool (http://www.wudapt.org), trying to develop useful inter-comparable time series of climate-relevant land use changes for urban settings around the world, starts from Landsat data.

Apologies for long-ish discussion of utility of Google Earth images.  In general, not relevant to ESSD users.

Direct comments on the manuscript, overall not up to standards expected for most journals.

Why the distinct asymmetry in corporate recognition?  A reader sees 'Google' constantly but 'Microsoft' almost never?

The data table downloads easily and opens in Excel, Google spreadsheet, Numbers, etc.  Authors used a 1 degree search algorithm.  Global 1 degree is 360 by 180 but assume no data at latitudes poleward of 80N and 60S so 50,400 possible data points, assume 30% land, gives 15,120.  Data table has 20 header rows followed by 15368 rows of lon lat data.  Assuming many interior data voids (northern Canada, Siberia, Greenland) combined with substantial overlap of Google Earth and Microsoft Bing in data rich regions, 15,000 rows of data seems about right?  But header (row 19) lists 59,168 data points.  At each of 15,000 lon lat points, Google Earth presence absence, Microsoft Bing presence absence, and date of most recent Google Earth scene, gives about 60,000 values?

Too many of these types of punctuation errors: "Bing Maps(Fritz et al.,."  Occurs due to intersection errors between reference software and word processor.  Authors should have searched and fixed these beforehand.

Page 2, line 5: LANDSAT operates jointly by NASA and USGS. Most researchers interact with USGS because they manage data distribution.  Present fuss about charging again for Landsat images - a spectacularly bad idea possibly fatal for the use of Landsat products in earth system research - centres on US Dept of Interior and USGS.

Page 2, line 6: "see e.g.(Microsoft, 2017)". Should instead read as '(see e.g. Microsoft, 2017)'. This error occurs in several places; authors should have made effort to search and correct.

Page 2 line 7: "still covered by Landsat resolution imagery, i.e. 15 m when pan-sharpened." Because Landsat resolution has evolved both in sensor resolution and data availability, not clear to readers which Landsat resolution the authors reference here?  Same text and same problem on Page 17 line 15.  Most recent Landsat 8 visible at 30 metres or panchromatic at 15 metres?

Page 2 line 21: the demise of the Google Earth API/plugin occurred earlier for some browsers? Google announced it as early as 2014 or 2015?  Do the authors address the issue of tracking image time series from this point forward, e.g. following the loss of the API tool?

Page 6 Figure 3.  Potentially useful figure but colour scheme detracts?  Very hard to distinguish Google only from Bing only, and which most recent.  Probably not accessible to colour-blind users.  Consider a different colour scheme with much higher contrast?  Category Google only also represents Google more recent by default? Likewise for Bing?  Very difficult to view and accept the authors' conclusions about Australia, for example, from this plot.  This reader estimates perhaps 60% Bing vs 40% Google for Australia, but with Bing predominant in the central outback while Google coverage dominates the agricultural and urban coastal regions?  Figure 3 does not seem to support the text sentence about relative lack of imagery in the Amazon basin or in Australia?  High latitudes and Sahara/Sahel yes, but not Amazon or Australia?   Why the apparent data hole over Afghanistan?  Why the abrupt discontinuities at US-Canada, India-China and Brazil-Bolivia borders?  Authors have avoided obvious features while discussing minor features in Australia or Indonesia?

Need a systematic approach, to show careful (as opposed to apparently random) analysis by authors and to enhance utility to users.  Start by latitude band?  Then move to terrestrial biomes to replace ad hoc mention of e.g. 'temperate' or 'deserts' or 'northern high latitudes'.  Because the narrative lacks organisation and structure, a reader can't distinguish useful from non-useful.  Abundant in one place relates to absence in another.

Page 7 line 1: here we read about relative abundance of imagery for Australia whereas on Page 5 we read about a lack of imagery for Australia?  Weakness in either the language or the analysis?  All these 'conclusions' could change if authors presented data in an area-conservative map projection?

Page 8,9, Table 1: possibly useful, but here we find, for example, 70% and 100% coverage for Australia (Google vs Bing) and approximately 70% and 90% for the authors' category "Most of South America".  Again this apparent mis-match between what a user reads in the text vs what the user finds in the maps or tables?   Perhaps the authors need to define their terms for abundant or deficient?

Eastern Europe shows by far the worst coverage (but gets relatively little attention in the text?), presumably because by these definitions Eastern Europe includes high-latitude Siberia?  We would learn more from a comparison of coverage by latitude, at least in the northern hemisphere, than from a coverage by geopolitical region?

Page 10, Figure 4, here "parts of Eastern Europe" qualify as "areas with the most imagery available".  Authors should adhere to a careful scale of most, many, abundant, few, etc.  Too much confusion and apparent discretion.

Apparently, researchers can access relatively abundant imagery for "some of the more populated regions across all the continents" but at the same time will find modest to low correlations of numbers of images with population in the least populated places with "no correlations in the rest of the world".   Authors have raised but not resolved a contradiction here: most VHR scenes available for populated areas but at the same time no correlation between scenes and population centres?

Page 12, Table 2: Protected area relevance would make much more sense on an areal basis rather than the presence-absence approach given here?  E.g. number of images that provide extensive coverage per area of protected region by geographic region?  A large number of images concentrated in a relatively small protected area have less impact than a few images across a large area?  If Eastern Europe or eastern US have relatively large numbers of images but relatively small areas of protection, those regions will distort or invalidate this analysis?  Overall, with 3 or fewer images per protected area location, this entire topical discussion seems moot?

Page 13, deforestation: This sentence does not make sense: "There is good coverage by Bing Maps in the Amazon and the Congo basin but there is only one image available and the most recent, frequent year found is 4 to 6 years old."  One image constitutes "good coverage"?  Due to this confusion, the following sentence about contrast results from Google also makes no sense.

Page 14, cropland: Again, this sentence makes little sense: "The results show that the cropland areas in these countries are covered by more than 90% VHR imagery in Google Earth; there are similar findings in Bing Maps except for Nigeria and Indonesia, which still have high coverage." What means "high" relative to 90%? Very confusing!

Page 15, 16, Table 4: all countries except Mongolia have greater than 90% and 6 (Google) or 8 (Bing) have 100% coverage. Percentage differences come down to presence or absence of 1 image! Too much inference based on too little information content?

Pages 5, 17 and Table 5: comparison with urban areas. Authors have earlier pointed out the absence of correlation of image numbers with population but here users get a sense of positive correlation with urban areas. ?? Most researchers access current population data from CIESIN (Center for International Earth Science Information Network, Gridded Population of the World, Version 4 (GPWv4): https://doi.org/10.7927/H4PG1PPM). The so-called JRC layer as cited provides a gridded version of GPWv4 but in a spatial raster format less useful to many users.

Page 17 and 18: discussion. The points raised in discussion about the abundance of VHR imagery and the potential utility of that imagery seem valid, but in too many places apparently inconsistent with earlier text among the results.

Examples -

a) If the authors mention the northern parts of Columbia or Ecuador, or parts of Indonesia (which confusingly, shows an imagery deficit on page 5 line 16 but an imagery abundance on Page 10 line 3 and Page 14 line 9), then we should also get some discussion of Afghanistan?
b) This combination of sentences and text does not make sense: "In the rest of the world there is some complementarity between Google Earth and Bing Maps, e.g. there are only Bing Maps present in parts of Canada, the Amazon, former Soviet Union countries and parts of Australia where Google Earth has no coverage. In contrast, Google Earth imagery adds very little additional spatial coverage …" What "complementarity"?
c) "the amount of historical imagery is actually quite small" (I agree!) but earlier we read (Page 7) that "North America, Southern Europe, Southern Africa, and Southern and Southeastern Asia have the richest archive of images". "Rich archive" vs "quite small"? How does a user / reader know how to judge this information? Where should they look for useful imagery?
d) "availability of VHR imagery in protected areas was surprisingly poor in North America, Eastern Europe and South America, particularly in Google Earth within the latter two regions" but Table 2 shows all regions except Eastern Europe above 50% image presence but in most cases only 3 images per area. Confusing?

The reference list seems very weak. It consists predominantly of reports, AGU abstracts, and self-promotional database or data portal documents. I count only 5 or 6 valid scientific publications using VHR imagery. The authors tend to defeat their case with this clearly-padded list.

If VHR scenes become both more available and more useful, readers will need a much more organised, systematic and compelling guide than the one provided here. At best, it seems pre-mature and not up to the quality expected for ESSD.

---

## Short Comment (SC2) · 5 Jul 2018

This is an interesting study that could be potentially very useful. Despite that the amount of earth observation data is increasing and the most data are processed automatically to create useful products, quality assessments and calibration of such earth observation product heavily rely on (mostly manual) interpretation of VHR satellite data.

In activities of collecting calibration and validation data, information on available VHR of different providers/platforms (Google and Bing map) is extremely valuable. There is limited literature available on the availability of open access VHR data. Such information should be made available (1) to the readers to take into consideration which

platform (in which regions) to use and (2) to raise awareness on the importance of access to VHR data for research communities and data providers.

It would be nice to have a summary of applications that require information on VHR data (what kind of applications/literature use (which) VHR data for what purposes). For example, large-scale land cover mapping often uses VHR data for reference data collection. Collect Earth platform allows visualization of VHR data mostly for forest change monitoring purposes. Emphasis on this "demand" would make it more appealing.

---

## Short Comment (SC3) · 13 Jul 2018

[revised manuscript text omitted]

---

## Author Comment (AC1) · 17 Jul 2018

Thank you for the review of this manuscript. Overall, we think you may have misunderstood the aim of the paper. We are not seeking to provide a full list of metadata such as that available from Landsat but a critical overview of the global availability of very high resolution (VHR) imagery in Google Earth and Microsoft Bing. The main advantage of VHR scenes is the level of spatial detail that one can detect in comparison to Landsat and Sentinel imagery. Because of this advantage, researchers can use VHR scenes to collect calibration and validation data through visual interpretation for use in remote sensing applications, and more recently, researchers have used VHR imagery for implementing statistical surveys for monitoring land features. In particular, the data set shared in this paper can be used to develop sampling designs for different monitoring tasks.

On this basis, we disagree with your two key points as explained in more detail below.

(1) Earth sciences researchers, at least those that publish wide ranges of data in ESSD, use imagery from Google Earth rarely (see below) and from Microsoft Bing almost never. This manuscript, which counts availability of Google Earth and Microsoft Bing scenes at various terrestrial locations of our planet, has minimal utility and relevance for ESSD readers. Separate from irrelevance, it fails substantially in the quality of its presentation.

If this paper is really not of interest to readers of ESSD, then we find it intriguing that over a very short period of time, this paper has received more views, and in particular downloads, than many other papers submitted to ESSD. We are sure that the Editor has access to the full statistics and could easily exclude views and downloads that were recorded from Austria, where we are based, and the Editor could compare these numbers to other publications in ESSD. The Editor could also examine these stats prior to the posting of your review. We consider the number of downloads as a pretty fair indication of interest.

From a more scientific content point of view, we would argue that there is a growing interest in the use of VHR imagery from Google Earth and Microsoft Bing but little is known about what is currently available temporally and spatially. This is one of the first papers that has addressed this issue. We feel that the Reviewer may have misunderstood the idea of the paper, i.e. it is not to replace other open satellite data like Copernicus or Landsat but to understand where validation and calibration data are available to train algorithms, which can then be used to classify Landsat or Sentinel type data or for visual interpretation of VHR to produce a validation or statistical sample.

A very recent paper by Bastin et al. (2017) on calculating global statistics of dry forests

(published in Science - DOI: 10.1126/science.aam6527 with 60 citations already in Google Scholar) has used Google Earth and Microsoft Bing imagery to create an independent statistical assessment that would not have been possible without this freely available VHR imagery. Moreover, both of these sources of imagery have been used to collect validation and calibration data for the production of the majority of global land cover maps, e. g. Hansen's forest cover maps, ESA CCI LC 20m, Copernicus LC time series at 100m, DLR's Global Urban Footprint, etc. although the fact that these data have been collected through visual interpretation of Google Earth and Microsoft Bing imagery is not always stated explicitly.

One might argue that such a paper may be more appropriate for a Remote Sensing journal. However, in the paper we show that these data can be of value for different monitoring applications. ESSD is an interdisciplinary journal that covers different research topics, and this paper covers a number of application areas including geosciences, agronomy, biodiversity, etc. Hence we feel the paper is of broader interest to researchers beyond remote sensing alone.

If we understood it correctly, the reasons why the Reviewer considers that the presentation of the paper is not comprehensive are given in the direct comments to the paper. These comments are addressable/manageable (see attachment) and we will take these comments into account during the manuscript revision.

(2) If VHR scenes become more available and more useful, readers will need a much more organized, systematic and compelling guide than the one provided here. At best, it seems pre-mature and not up to the quality expected for ESSD.

We will address this comment based on three aspects: availability; usefulness; and quality of the analysis.

Availability: This review paper shows that at least 64% and 70% of global land covered by VHR is available for visual interpretation in Google Earth and Microsoft Bing, respectively. Moreover we show where these scenes are available and what the dates

are. However, the Reviewer seems to see only the data gaps and focuses only on the limitations, some of which we actually discuss in the paper. We want to raise the point that exactly in those 70% of Earth's land surfaces, researchers can benefit considerably from VHR, which is available (for free) in Google Earth and Microsoft Bing. We agree that only having visual information available and not the full spectral bands is indeed limiting yet we would argue that many landscape-related and environmental monitoring applications can be undertaken using this visual information. Moreover, we observed that the main data gaps are in the northern latitudes and in areas with high precipitation where all optical sensors suffer from a lack of data, including Sentinel-2 and Landsat. We also show that there are places where there are three images available after the year 2010, which is particularly important for studies on change detection. Note that Sentinel-2 imagery is only available from 2015 onwards.

Usefulness: The main advantage of the VHR imagery reviewed in this paper is that users can visually detect much more detail about land cover than is visible from Sentinel and Landsat data; please see Figure 1 for an example that illustrates this point clearly. For example, individual trees, shrub shelterbelts, field boundaries, etc., are clearly visible from VHR imagery. This ability to distinguish these features is what makes the images extremely useful for a range of applications, e.g. such as those provided in the manuscript. We agree that there are certain limitations to the analysis, e.g. open and straightforward access to the dates in Google Earth is no longer possible with the closure of the Google Earth API, there are issues related to the positional accuracy of VHR scenes, the resolution changes when going back into the past, etc. However, from our perspective and for the range of applications discussed in the paper, the most important information is whether VHR imagery is present or not and if so, for which season, how many scenes, how frequent they are, how far they go back into the past, etc. This is the type of information that is made available through this review paper. Other data such as the viewing angle is currently not available but is less important since objects, e.g. the presence or absence of trees, can be visually detected. Viewing angle is also less important if the georeferencing has been done well. Again, in

the discussion, the Reviewer focuses only on missing information, which is not always important for the data purposes that we discuss in the paper.

Quality of the analysis: For the analysis, we used a data set that consisted of circa 15 K points using a systematic sample with 1 degree spacing. This is a sufficient number of samples to provide a global snapshot of VHR availability for readers. We acknowledge that this is not a large enough sample for doing very detailed analysis of small area objects, e.g. protected areas at the national level. However, this exploratory exercise will guide readers/users in a making a decision regarding whether to use VHR imagery in their work for a specific area of interest, and also in developing a sampling design. Or they could simply use this paper as an example and then interrogate the Bing API to create a more detailed grid of dates as a first pass. There are also ways to interrogate Google Earth, which could be implemented to extract the information needed at a finer resolution. If sufficient VHR imagery were found to be available, this could then become the subject of a separate paper, e.g. monitoring protected areas in the US. Users would still need to do more detailed studies but they could use tools such as Collect Earth, LACO-Wiki or create their own bespoke application.

We believe that if there were more users of VHR imagery and consequently more published research articles that use the imagery available in Google Earth and Microsoft Bing, then a new API for accessing the dates of imagery in Google Earth as well as other meta-information about the satellite imagery should be provided to researchers. This paper aims to draw attention to this gap while raising awareness of what VHR imagery is currently available and how it might be used.

Only with time will we be able to evaluate the impact of this review paper but we disagree that such a paper should not be considered for publication. This paper opens up an area of research that is clearly of interest to many but is still currently under scientific investigation.

Please also note the supplement to this comment:

[Figure]

https://www.earth-syst-sci-data-discuss.net/essd-2018-13/essd-2018-13-AC1-supplement.pdf

[Figure]

[Figure]

**Fig. 1.** Illustration of the level of detail that users can detect. Figure on the left: a VHR image from Google Maps. Figure on the right: a Sentinel-2 image (natural colors) at 10 m resolution.

**Supplement:**

**The Reviewer's direct comments to the manuscript:**

Note: Here we have skipped the Reviewer's discussion regarding why this paper is not relevant since we already provided our arguments in the main response.

| Comment | Our response |
|---|---|
| Why the distinct asymmetry in corporate recognition? A reader sees 'Google' constantly but 'Microsoft' almost never? | We will correct this by inserting "Microsoft" in the appropriate places. |
| The data table downloads easily and opens in Excel, Google spreadsheet, Numbers, etc. Authors used a 1 degree search algorithm. Global 1 degree is 360 by 180 but assume no data at latitudes poleward of 80N and 60S so 50,400 possible data points, assume 30% land, gives 15,120. Data table has 20 header rows followed by 15368 rows of lon lat data. Assuming many interior data voids (northern Canada, Siberia, Greenland) combined with substantial overlap of Google Earth and Microsoft Bing in data rich regions, 15,000 rows of data seems about right? But header (row 19) lists 59,168 data points. At each of 15,000 lon lat points, Google Earth presence absence, Microsoft Bing presence absence, and date of most recent Google Earth scene, gives about 60,000 values? | Thank you for this observation. We double checked the number of records and there are a few missing rows. We will update the data set on Pangaea. |
| Too many of these types of punctuation errors: "Bing Maps (Fritz et al.,." Occurs due to intersection errors between reference software and word processor. Authors should have searched and fixed these beforehand. | We will correct these typos. |
| Page 2, line 5: LANDSAT operates jointly by NASA and USGS. Most researchers interact with USGS because they manage data distribution. Present fuss about charging again for Landsat images - a spectacularly bad idea possibly fatal for the use of Landsat products in earth system research - centres on US Dept of Interior and USGS. | We will add this point to the discussion. |
| Page 2 line 7: "still covered by Landsat resolution imagery, i.e. 15 m when pan-sharpened." Because Landsat resolution has evolved both in sensor resolution and data availability, not clear to readers which Landsat resolution the authors reference here? Same text and same problem on Page 17 line 15. Most recent Landsat 8 visible at 30 metres or panchromatic at 15 metres? | Thank you. We will correct this in the text. It should be 15 meter resolution visible. |
| Page 2 line 21: the demise of the Google Earth API/plugin occurred earlier for some browsers? Google announced it as early as 2014 or 2015? Do the authors address the issue of tracking image time series from this point forward, e.g. following the loss of the API tool? | We will add this limitation to the discussion along with potential solutions. |
| Page 6 Figure 3. Potentially useful figure but colour scheme detracts? Very hard to distinguish Google only from Bing only, and which most recent. Probably not accessible to colour-blind users. Consider a different colour scheme with much higher contrast? Category Google only also represents Google more recent by | We tried a number of different color schemes and chose what we felt was the most contrasting. Regarding color blindness, we did not use red and green |

| | |
|---|---|
| default? Likewise for Bing?  Very difficult to view and accept the authors' conclusions about Australia, for example, from this plot.  This reader estimates perhaps 60% Bing vs 40% Google for Australia, but with Bing predominant in the central outback while Google coverage dominates the agricultural and urban coastal regions? Figure 3 does not seem to support the text sentence about relative lack of imagery in the Amazon basin or in Australia?  High latitudes and Sahara/Sahel yes, but not Amazon or Australia?   Why the apparent data hole over Afghanistan?  Why the abrupt discontinuities at US-Canada, India-China and Brazil-Bolivia borders?  Authors have avoided obvious features while discussing minor features in Australia or Indonesia?

Need a systematic approach, to show careful (as opposed to apparently random) analysis by authors and to enhance utility to users.  Start by latitude band?  Then move to terrestrial biomes to replace ad hoc mention of e.g. 'temperate' or 'deserts' or 'northern high latitudes'. Because the narrative lacks organization and structure, a reader can't distinguish useful from non-useful.  Abundant in one place relates to absence in another. | colors next to each other. Instead of red, we used a brown color. However, we will revisit this and attempt to improve the color schemes in the revised version.

Thanks for these comments. We will take a more systematic approach to the way we describe the findings and improve the presentation in the revised version.

We will pay more attention to the use of words, e.g. abundant vs absence so that these are not contradictory in the revised version. |
| Page 7 line 1: here we read about relative abundance of imagery for Australia whereas on Page 5 we read about a lack of imagery for Australia?  Weakness in either the language or the analysis?  All these 'conclusions' could change if authors presented data in an area-conservative map projection? | We will change the words to be consistent throughout the text. We used Robinson projection for visualization of the results. We will check if the conclusions will change if we use e.g. Goode homolosine projection, we will provide examples. |
| Page 8,9, Table 1: possibly useful, but here we find, for example, 70% and 100% coverage for Australia (Google vs Bing) and approximately 70% and 90% for the authors' category "Most of South America".  Again this apparent mis-match between what a user reads in the text vs what the user finds in the maps or tables? Perhaps the authors need to define their terms for abundant or deficient? | This comment is related to those you raised above. We will use descriptive words more consistently throughout the text in the revised version. |
| Eastern Europe shows by far the worst coverage (but gets relatively little attention in the text?), presumably because by these definitions Eastern Europe includes high-latitude Siberia?  We would learn more from a comparison of coverage by latitude, at least in the northern hemisphere, than from a coverage by geopolitical region? | We believe that most of the readers do their research by countries or by world regions rather than by latitude. Hence the practical value of this analysis is unclear for us. However, as mentioned in comments above, we will take a more systematic approach to the comparison in the revised version. |
| Page 10, Figure 4, here "parts of Eastern Europe" qualify as "areas with the most imagery available".  Authors should adhere to a careful scale of most, many, abundant, few, etc.  Too much confusion and apparent discretion. | As mentioned in the comments above, we will adjust the language to provide a better |

| | |
|---|---|
| Apparently, researchers can access relatively abundant imagery for "some of the more populated regions across all the continents" but at the same time will find modest to low correlations of numbers of images with population in the least populated places with "no correlations in the rest of the world". Authors have raised but not resolved a contradiction here: most VHR scenes available for populated areas but at the same time no correlation between scenes and population centres? | characterization of what we mean by terms such as most, many, abundant, etc. We will also revise our statements regarding the correlation with population centres. |
| Page 12, Table 2: Protected area relevance would make much more sense on an areal basis rather than the presence-absence approach given here? E.g. number of images that provide extensive coverage per area of protected region by geographic region? A large number of images concentrated in a relatively small protected area have less impact than a few images across a large area? If Eastern Europe or eastern US have relatively large numbers of images but relatively small areas of protection, those regions will distort or invalidate this analysis? Overall, with 3 or fewer images per protected area location, this entire topical discussion seems moot? | With this example, we want to show that VHR imagery could potentially be used for monitoring protected areas. A more detailed study on this topic (or one that focuses on specific protected areas) would be required and is a separate paper. |
| Page 13, deforestation: This sentence does not make sense: "There is good coverage by Bing Maps in the Amazon and the Congo basin but there is only one image available and the most recent, frequent year found is 4 to 6 years old." One image constitutes "good coverage"? Due to this confusion, the following sentence about contrast results from Google also makes no sense. | We will rephrase this sentence in the revised manuscript. |
| Page 14, cropland: Again, this sentence makes little sense: "The results show that the cropland areas in these countries are covered by more than 90% VHR imagery in Google Earth; there are similar findings in Bing Maps except for Nigeria and Indonesia, which still have high coverage." What means "high" relative to 90%? Very confusing! | We will amend the text to better explain this point. |
| Page 15, 16, Table 4: all countries except Mongolia have greater than 90% and 6 (Google) or 8 (Bing) have 100% coverage. Percentage differences come down to presence or absence of 1 image! Too much inference based on too little information content? | The aim of Table 4 is not to compare the percentages for these few countries. Rather the idea is to provide a quick overview of the availability of VHR imagery for anyone that is interested in monitoring cropland in these countries. |
| Pages 5, 17 and Table 5: comparison with urban areas. Authors have earlier pointed out the absence of correlation of image numbers with population but here users get a sense of positive correlation with urban areas. ?? Most researchers access current population data from CIESIN (Center for International Earth Science Information Network, Gridded Population of the World, Version 4 (GPWv4): https://doi.org/10.7927/H4PG1PPM ). The so-called JRC layer as cited provides a gridded version of GPWv4 but in a spatial raster format less useful to many users. | The Reviewer has misunderstood the meaning of Table 5. It aims to answer the question regarding whether VHR imagery can be used for monitoring urban areas. As mentioned above, we will address the comment about the correlation with urban centres. |

| | We can use the CIESIN data instead of the JRC layer to see if this changes the analysis. |
|---|---|
| Page 17 and 18: discussion. The points raised in discussion about the abundance of VHR imagery and the potential utility of that imagery seem valid, but in too many places apparently inconsistent with earlier text among the results.  Examples -
a)If the authors mention the northern parts of Columbia or Ecuador, or parts of Indonesia (which confusingly, shows an imagery deficit on page 5 line 16 but an imagery abundance on Page 10 line 3 and Page 14 line 9), then we should also get some discussion of Afghanistan?
b)This combination of sentences and text does not make sense: "In the rest of the world there is some complementarity between Google Earth and Bing Maps, e.g. there are only Bing Maps present in parts of Canada, the Amazon, former Soviet Union countries and parts of Australia where Google Earth has no coverage. In contrast, Google Earth imagery adds very little additional spatial coverage ..." What "complementarity"?
c)"the amount of historical imagery is actually quite small" (I agree!) but earlier we read (Page 7) that "North America, Southern Europe, Southern Africa, and Southern and Southeastern Asia have the richest archive of images". "Rich archive" vs "quite small"?  How does a user / reader know how to judge this information?  Where should they look for useful imagery?
d)"availability of VHR imagery in protected areas was surprisingly poor in North America, Eastern Europe and South America, particularly in Google Earth within the latter two regions" but Table 2 shows all regions except Eastern Europe above 50% image presence but in most cases only 3 images per area.  Confusing? | We will address these inconsistencies (in particular those pointed out in comments a to d) as well as any others in the revised version of the manuscript. |
| The reference list seems very weak. It consists predominantly of reports, AGU abstracts, and self-promotional database or data portal documents. I count only 5 or 6 valid scientific publications using VHR imagery. The authors tend to defeat their case with this clearly-padded list. | All the references in which we are authors are relevant to the subject of this paper and hence are justified in being cited. There are not that many other references that use VHR imagery, which the Reviewer has also raised. Since the paper has been submitted, there have been a few more publications that have appeared; we will add these to references. |

---

## Short Comment (SC4) · 19 Jul 2018

Very high-resolution (VHR) imagery available via online mapping platforms, such as Google EarthTM are commonly utilized for various applications, including the collection of training and validation data for land-cover classifications. However, heavy criticism has been placed on such studies due to a greater spatial and temporal variability of VHR imagery. Despite existing attempts to synthesize the records of VHR imagery across publically-available domains, this is the first, to my successful knowledge implementation of such spatial analysis and synthesis of VHR products.

The advantage of this study stems from an analytical perspective but also making

analysis and evolving products publically available. This would certainly allow fine-tuning the selection of training and validation data, but also to think carefully about large areas that are sliding off from current VHR coverage.

Unfortunately, though, even this product is timing, in a few years, the relevance of this work will be probably diminished due to changing paradigm in data accessibility and appearance of new products, including microsatellite constellations. On thematic analysis, I was a bit curios that probably not only population density determines interest in recording and making them publically accessible VHR satellite footprints. It seems, the income levels, developed versus, transition and developing countries, might also determine the availability of such VHR imagery (example, the concentration of observations across the US and the EU). This might be underlined in the paper, since it may guide the policies and different levels of accessibility for such datasets.

In sum, the manuscript is timing, with simple but neat spatial analysis and is highly welcomed. It is even more appreciated because authors considered to share produced datasets. Well done.